# Deciphering the Proteome and Phosphoproteome of Peanut (*Arachis hypogaea* L.) Pegs Penetrating into the Soil

**DOI:** 10.3390/ijms26020634

**Published:** 2025-01-14

**Authors:** Sha Yang, Mei He, Zhaohui Tang, Keke Liu, Jianguo Wang, Li Cui, Feng Guo, Ping Liu, Jialei Zhang, Shubo Wan

**Affiliations:** 1Institute of Crop Germplasm Resources, Shandong Academy of Agricultural Sciences, Jinan 250100, China; yangsha@shandong.cn (S.Y.); hemei920@163.com (M.H.); tangzhaohui1116@163.com (Z.T.); 13606413632@163.com (K.L.); wang_jianguo2020@163.com (J.W.); cuili0557@126.com (L.C.); gf123456gh@126.com (F.G.); 2Shandong Academy of Agricultural Sciences Institute of Agricultural Resources and Environment, Jinan 250100, China; liuapple5326@sina.com

**Keywords:** peanut (*Arachis hypogaea* L.), pegs, pod development, protein phosphorylation, stress resistance

## Abstract

Peanut (*Arachis hypogaea* L.) is one of the most important crops for oil and protein production. The unique characteristic of peanut is geocarpy, which means that it blooms aerially and the peanut gynophores (pegs) penetrate into the soil, driving the fruit underground. In order to fully understand this phenomenon, we investigated the dynamic proteomic and phosphoproteomic profiling of the pegs aerially and underground in this study. A total of 6859 proteins and 4142 unique phosphoproteins with 10,070 phosphosites were identified. The data were validated and quantified using samples randomly selected from arial pegs (APs) and underground pegs (UPs) by parallel reaction monitoring (PRM). Function analyses of differentially abundant proteins (DAPs) and differentially regulated phosphoproteins (DRPPs) exhibited that they were mainly related to stress response, photosynthesis, and substance metabolism. Once the pegs successfully entered the soil, disease-resistant and stress response proteins, such as glutathione *S*-transferase, peroxidase, and cytochrome P450, significantly increased in the UP samples in order to adapt to the new soil environment. The increased abundance of photosynthesis-associated proteins in the UP samples provided more abundant photosynthetic products, which provided the preparation for subsequent pod development. Phosphoproteomics reveals the regulatory network of the synthesis of nutrients such as starch, protein, and fatty acid (FA). These results provide new insights into the mechanism, indicating that after the pegs are inserted into the soil, phosphorylation is involved in the rapid elongation of the pegs, accompanied by supplying energy for pod development and preparing for the synthesis of metabolites during pod development following mechanical stimulation and darkness.

## 1. Introduction

Peanut (*Arachis hypogaea* L.) is a major oil and economic crop and rich in edible fats and proteins [1]. Among legumes, peanuts are particularly unusual for their geocarpy. Peanut plants produce aerial flowers, and once the ovules are fertilized, the meristem cells of the ovary divide rapidly so that the ovary stalk is constantly extended. This elongated ovary stalk containing the developing seeds that grows out of the withered calyx tube is the gynophore, which, in peanuts, is usually referred to the peg [2]. Fertilized ovaries are sent underground as the pegs grow into the soil, and then the pods develop and mature. Understanding the mechanism of peanut peg geotropism is important for exploring the biological basis of peanut pod development and thus yield formation.

Some studies severed the ovary at the tip of the pegs prior to it being placed into the soil; the remaining ovary stalk stopped stretching and could not be pushed into soil as normally occurs [3]. Thus, normal fertilization and development thereafter were observed to be necessary for the formation and elongation of the gynophores into the soil. Furthermore, darkness and mechanical stress are crucial factors for peg penetration and pod formation [4]. The geotropism of the peanut gynophores encourages their growth into the soil to ensure pod growth. Once the pegs have successfully entered the soil, they will be transferred from aerial growth to a new growing environment—soil. Soil contains a variety of granular minerals, water, air, organic matter, and microorganisms. Abiotic stress may occur under these conditions owing to improper tillage and poor soil conditions, which will hinder the formation of full peanut pods and limit the resulting yield [5]. In addition, ongoing climate change is expected to profoundly impact both land use and abiotic factors, which will in turn alter plant–microorganism interactions in soils, including those involving soil-borne pathogens [6]. These pathogens often cause soil-borne diseases, which frequently remain undiagnosed and unmanaged, leading to chronic yield and quality losses as well. Consequently, the soil environment holds significant importance for the formation of peanut pods.

As an oil crop, peanut pod formation is also related to plant oil synthesis and accumulation. The main components of vegetable oil are fatty acids. During seed development, the biosynthesis of fatty acids is accomplished via multi-enzyme complex catalysis.

Acetyl-coenzyme A (acetyl-CoA) carboxylase catalyzes acetyl CoA to form malonyl CoA [7]. After condensation, reduction, and dehydration, malonyl-CoA and acetyl-CoA are catalyzed to synthesize fatty acids by fatty acid catalase. Through multiple cycles including reduction, two carbon atoms are added during each cycle to ultimately produce a 12–18-carbon fatty acid [8]. As a product of photosynthesis, sugars are transported from the leaves to the developing embryos, some of which directly enter the starch metabolic pathway and some of which go through the glycolysis pathway to produce pyruvate and then acetyl-CoA, which could be used as a substrate for fatty acid synthesis [9,10]. There are also enzymes involved in the degradation of lipids, such as lipoxygenase, phospholipase, and GDSL esterase/lipase proteins (GELPs), and some proteins involved in lipid β oxidation that are used to break down lipids can be used to meet the material and energy requirements for crop growth and morphogenesis [11].

In the past decade, various omics tools used in peanut research have emerged. Hundreds of genes involved in geocarpy were identified by transcriptomics [12]. The differentially expressed proteins of peanut gynophores at different developmental stages were also identified by proteomics [13]. Protein phosphorylation pertains to the transfer of a phosphate group from ATP/GTP to amino acid residues of proteins, which is catalyzed by protein kinases [14,15]. It constitutes one of the most significant post-translational modifications (PTMs) that can precisely regulate the function of proteins. Phosphorylation is reversible, and phosphorylated proteins can be dephosphorylated by phosphatases [16]. Protein phosphorylation and dephosphorylation play a crucial role in regulating biological processes such as cell proliferation [17], energy metabolism [18], and signal transduction [19,20]. Here, we have characterized, for the first time, the post-translational regulation of peanut gynophores before and after soil penetration by phosphoproteomics approaches. Our results offer valuable resources for investigating the development of peanut gynophores and geotropism in peanut.

## 2. Results

### 2.1. Phenotype Characteristics of Aerial and Underground Pegs

The pegs from early and intermediate flowers on the lateral branches of the base are easily buried in the soil, while the pegs from flowers on the middle and upper lateral branches are longer and gradually wilt because they are not easily pushed into the soil. As depicted in Figure 1A, we employed the peanut gynophores about to be inserted into the soil and the underground peanut gynophores that had just been inserted into the soil. (Figure 1A). The embryo and endosperm nuclei basically stop dividing at the initial stage of the insertion of the peg into the soil, and the embryonic development resumes after the peg stops elongating underground. Therefore, the observed lengths of the underground pegs varied significantly relative to the aerial pegs, while there was no obvious difference in the embryo specifically. A cross-cutting diagram of different areas of the peanut gynophores, such as the elongation zone, meristematic zone, and ovary, is shown in Figure 1B.

### 2.2. Proteomic Identification of Underground Pegs

Data of the proteome were accessible via Proteome Xchange with the identifier PXD042065. In this study, the total spectral number is 659,649, of which the matched spectral number is 293,262. The number of identified peptides is 37,020, and the number of identified proteins is 6859. For biologically replicated samples, Principal Component Analysis (PCA) was used to test whether the quantitative results of the biologically replicated samples conformed to statistical consistency (Figure 2A). The PCA revealed that the biological replicates of each sample clustered together in distinct areas, indicating that the reproducibility within each group was high, while the differentiation among groups was also high. In the proteome, a total of 6859 proteins were identified, encompassing 1160 up-regulated proteins and 808 down-regulated proteins (Supplement Appendix A and Appendix A). The protein abundance results were filtered according to a threshold of expression fold change (FC) of >1.5 in protein abundance and *p* < 0.05. Hence, 333 DAPs were identified in the UP compared with the AP samples, among which 210 were up-regulated and 123 down-regulated. (Figure 2B).

For enrichment analysis, DAPs were divided into four clusters (Q1–Q4) based on the fold change, such that Q1, Q2, Q3, and Q4 indicate FC < 0.5, 0.5 ≤ FC < 0.667, 1.5 < FC ≤ 2, and 2 < FC, respectively (Figure 2C,D). Protein domain enrichment analysis of peg DAPs in the proteome revealed enrichment of the pathogenesis-related protein Bet v I family, chitinase class I protein, and chitin recognition protein, which are involved in regulating plant disease resistance [21,22]. NADPH-dependent FMN reductase and the FAD binding domain play important roles in photosynthesis [23,24], and peroxidase and cytokinin dehydrogenase 1, which are involved in regulating plant growth [25], were noticeably up-regulated in UPs (Figure 2C). KEGG enrichment analysis indicated that the up-regulated DAPs were predominantly concentrated in tyrosine metabolism, isoquinoline alkaloid biosynthesis, phenylpropanoid biosynthesis, zeatin biosynthesis, and the MAPK signaling pathway (Figure 2D). It was observed that these up-regulated DAPs were predominantly implicated in the regulation of photosynthesis, the glycolytic pathway, and defense mechanisms.

### 2.3. Functional Classification and Verification of DAPs

We conducted an analysis of DAPs enriched in the COG (Clusters of Orthologous Groups of proteins) database. As shown in Figure 2E, (C) energy production and conversion, (E) amino acid transport and metabolism, (G) carbohydrate transport and metabolism, (O) post-translational modification, protein turnover, chaperone, and (Q) secondary metabolite biosynthesis and transport were the top five functional classes. These DAPs occur primarily in the cytoplasm, followed by the chloroplasts and nuclei (Figure 2F). As demonstrated in Appendix A, DAPs were also analyzed based on three classifications of GO: biological process (BP), cellular component (CC), and molecular function (MF). Within the cell component category, the DAPs identified in the UP and AP samples are mainly enriched in three principal terms: cell, intracellular, and protein-containing complexes. Catalytic activity and binding constitute the most abundant molecular functions. Metabolic processes, cellular processes, responses to stimulus, and biological regulation represent the four most significant categories of biological processes. The GO enrichment analysis of DAPs is presented in Appendix A.

The modified DAPs were compared with the STRING (v.11.0) protein interaction network database, and the protein interaction relationships were extracted according to a confidence score threshold > 0.7. In total, 152 nodes were assembled (Appendix A and Appendix A). Within these interaction networks, six subnetworks of significant enrichment were identified, including (1) flavonoid biosynthesis, (2) spliceosome, (3) cyanoamino acid metabolism, (4) protein processing in endoplasmic reticulum, (5) linoleic acid metabolism, and (6) SNARE interactions in vesicular transport (Appendix A). By analyzing the differential protein interaction network diagram, we were able to identify protein modules and pathways with similar functions and elucidate the regulatory mechanism of protein interactions, which helps to reveal the geotropic growth regulatory mechanisms of peanut gynophores.

### 2.4. Functional Classification and Verification of DRPPs

Data of the phosphorylated proteome were accessible via Proteome Xchange with the identifier PXD042066. In the differentiation region of the PCA map, the biological replicates of each sample were clustered together, indicating high intra-group consistency and inter-group differentiation, as well as high intra-group repeatability. (Figure 3A). A total of 4142 distinct phosphorylated proteins and 10,070 phosphorylation sites were detected. Among the diverse modification sites, 991 phosphorylation sites were up-regulated and 234 phosphorylation sites were down-regulated (Appendix A). The phosphorylation levels of the sites among the DRPPs were filtered according to a threshold value of *p* < 0.05, with increases and decreases of more than a 1.5-fold change considered significantly up- and down-regulated, respectively. In total, 289 phosphosites (243 increased and 46 decreased) corresponding to 225 (185 increased and 40 decreased) phosphoproteins were observed after the pegs had been pushed into the ground (Figure 3B). Notably, the majority of phosphorylated proteins possess only one phosphorylation site, while 948, 461, and 760 phosphorylated proteins contain two, three, or more phosphorylated sites, respectively. Among all the amino acids of identified phosphoproteins, serine, threonine, and tyrosine residues contained 8752, 1299, and 19 phosphorylation sites, respectively (Figure 3C).

Protein domain enrichment analysis of differentially phosphorylated proteins showed that cation transporter/ATPase, haloacid dehalogenase-like hydrolase, alpha/beta hydrolase fold, alpha/beta hydrolase fold domains, and lactate/malate dehydrogenase were all up-regulated (Figure 3D). KEGG enrichment analysis demonstrated that the up-regulated phosphorylated proteins were mainly enriched in metabolic pathways such as the proteasome, glyoxalate, and dicarboxylate metabolism (Figure 3E). Phosphorylation motif analysis between the AP and UP samples was performed. Ten motifs (i.e., RxxS, SP, TPR, SGPL, PxTP, SDDD, SDxD, SF, SPR, and PxSP) were overexpressed among phosphorylation sites, showing increased phosphorylation between the UP and AP samples (Figure 3F, Appendix A). RxxS motifs have been shown to be potential substrates for protein kinase A, mitogen-activated protein kinase (MAPKK), and calmodulin-dependent protein kinase (CDPK). The Pro-directed motifs SP and SPR, which are putatively phosphorylated by MAPKs, were found in 316 and 172 phosphorylated peptides, respectively [26,27]. Based on the signal transduction and response roles under stress of the proteins mentioned above, the conserved motifs appear to be essential to the abiotic stress signaling and response.

As shown in Figure 3G, the COG analysis of DRPPs indicated that (C) energy production and conversion, (O) post-translational modification, protein turnover, chaperone protein, and intracellular transport, (U) secretion and vesicular transport, (T) signal transduction mechanisms, and (Q) biosynthesis, transport, and catabolism of secondary metabolites were overrepresented as the main functional categories (Figure 3G). It is predicted that DRPPs are associated with the nucleus, chloroplast, cytoplasm, and plasma membrane (Figure 3H). Just as DAPs were analyzed for enrichment according to the three GO classifications (i.e., BP, CC, and MF), the DRPPs were also analyzed, as shown in Appendix A. Among the DRPPs, cellular process, metabolic process, and response to stimulus were enriched terms in the BP classification. The signaling, other, and reproduction terms were enriched in the CC classification. The enrichment items of MF mainly comprised binding activity, catalytic activity, and transporter activity. The GO enrichment analysis of DRPPs is shown in Appendix A.

The modified DRPP sites were compared with the STRING (v.11.0) protein interaction network database, and a threshold of confidence score > 0.7 was set to analyze the protein interaction relationship. In total, there were 352 nodes identified (Appendix A). These interaction networks identified six significant enrichment subnetworks, including protein processing in endoplasmic reticulum, SNARE interactions in vesicular transport, endocytosis, RNA transport, peroxisome, and starch and sucrose metabolism (Appendix A).

### 2.5. Significantly Differentiated Kinase Phosphorylation Sites

The perturbation of kinase activity plays an important role in mediating stress signaling and plant immunity, and the phosphorylation of kinases determines the kinase activity. For the identified phosphorylation modification sites, a dot diagram was drawn after sequencing according to log_2_(ratio) values. For the data shown in Figure 4A, phosphorylated proteins with their unique expression patterns are marked in red, and the protein accession_modification sites are marked with lines (also see Appendix A). These phosphorylated proteins included phosphoenolpyruvate carboxykinase, calcium-dependent protein kinases (CDPKs), and protein kinase.

### 2.6. Detection of Phosphorylated Sucrose-Phosphate Synthase and Verification of Omics Data

Phos-Tag molecules and a subsequent Western blot assay were utilized to assess the potential phosphorylation of sucrose phosphate synthetase (SPS) [28]. Figure 4B shows that the phosphorylation level of SPS, which is a key enzyme controlling sucrose synthesis, increased after the pegs had been pushed into the ground.

Parallel reaction monitoring (PRM) validation was combined with the quantification of peptides and peptide segments to complete the comprehensive analysis of proteins. To validate the accuracy of the proteomes and peptides using TMT-label-based quantitative analysis, 19 target proteins and 20 target modified peptides were selected for PRM protein targeting verification (Figure 4C,D). The protein expression in the PRM assay exhibited a similar trend to the quantitative results of TMT, which verifies the reliability of the TMT data.

### 2.7. The Expression of Some Significantly Enriched DAPs and Phosphorylation Level of DRPPs in APs and UPs

The induction of peroxidases, which are involved in the plant antioxidant defense process, helps plants maintain their physiological adaptability to the environment. Further analysis showed that a majority of 12 glycosyltransferases were uniformly up-regulated after the peg had been pushed into the ground (Figure 5A). In addition, eight peptidase proteins, which are involved in enhancing the resistance of plants to abiotic stress, were also up-regulated in the UP samples. Accordingly, eight lipase_GDSL proteins were down-regulated in the UP samples (Figure 5A). Further analysis showed that many protein kinases probably involved in glyconeogenesis and protein phosphorylation, including malate dehydrogenases, malic enzymes, sucrose-phosphate synthases, fructose-bisphosphate aldolases, and protein kinases, were up-regulated (Figure 5B). The contents of amylose and amylopectin increased in UP. Among them, the amylose content increased by 34.82%, while the amylopectin content increased by 17.51% (Figure 6).

## 3. Discussion

In recent years, a large number of different omics research methods have been broadly applied to investigate the geotropism of peanut gynophores, as reported in previous transcriptomic analyses [29,30], the *Arachis duranensis* genome draft, and proteome analysis [31]. In the present study, proteomic and phosphoproteomic methods were used to reveal the quantitative differences in protein and protein post-translation modification between aerial and subterranean peanut gynophores, providing new molecular insights into geocarpy in peanut. Thus, we identified 333 DAPs and 225 DRPPs, as well as 289 different phosphosites in the UP compared with the AP samples.

In the proteome, DAPs between UPs and APs were mainly enriched in the disease-related protein betvI family, chitinase class I, and chitin recognition proteins, indicating that the new environment induced plant disease resistance enhancement after pegs penetrate into the soil [21,22]. NADPH-dependent FMN reductase and the FAD binding domain, which play an important role in photosynthesis [24], and peroxidase and cytokinin dehydrogenase 1, which are involved in regulating plant growth, were significantly up-regulated in the UP samples (Figure 2C). Glucose-6-phosphate/phosphate and phosphoenolpyruvate/phosphate antiporter, ribulose bisphosphate carboxylase, photosystem II D2 protein, and phosphoenolpyruvate carboxykinase are closely related to the photosynthetic regulation and light signal transduction pathway, suggesting that photosynthetic potential increased as the pegs penetrated into the soil, which is conducive to the accumulation of photosynthetic products. Glucose-6-phosphate, phosphoenolpyruvate, and phosphoenolpyruvate carboxykinase are implicated in the glycolysis pathway. The metabolism of sugar is the central hub of all biological processes, linking protein, lipid, nucleic acid, and secondary substance metabolism [32]. Sugars synthesized through leaf photosynthesis are transported to storage organs via the phloem and unloaded into storage cells to provide energy for subsequent organ growth, fruiting, seed expansion, and ripening [33]. The metabolism and transport distribution of sugars also impact the composition of sugars entering storage cells and ultimately affect the quality of crops, including fruits and vegetables. The present findings suggested that the essential nutrients for pod development begin to accumulate during peg insertion, ensuring the full development of peanut pods. Aerial pegs are exposed to air, and when they grow underground, they come into contact with soil. Changes in the environment trigger the plant’s immune system, leading to an increase in the expression of disease-resistant and stress response proteins, such as glutathione S-transferase, cytochrome P450, and peroxidase, which were significantly increased in the UP samples.

Protein phosphorylation is a kind of protein post-translational modification, which is involved in the regulation of cytoplasmic enzyme activity [34,35]. For instance, phosphorylation has been shown to alter the activity of SPS and thus regulate the sucrose metabolism of *Zea mays* L. [36]. In our study, four phosphorylation sites of SPS, i.e., Ser-134, Ser-700, Ser-712, and Ser-715, were detected, and all four phosphorylation sites were significantly up-regulated in the UP samples (Figure 4B and Figure 5B). SPS serves as a pivotal regulatory node in the allocation of photosynthetic products towards sucrose and starch [37]. Therefore, its activity directly affects the distribution of photosynthates and is positively correlated with sucrose formation and negatively correlated with starch accumulation. After the pegs penetrated into the ground, SPS was phosphorylated and deactivated in the dark soil environment, indicating that sucrose is mainly transported through pegs rather than synthesized. Sucrose was transported from source to sink, which promotes the synthesis of starch (Figure 6).

Malate dehydrogenase (EC: 1.1.1.37, MDH) is one of the important enzymes in the tricarboxylic acid cycle (TCA), which can catalyze the reversible conversion of malic acid and oxaloacetic acid. This process relies on the NAD^+^/NADH cofactor system to complete the reaction and plays a vital role in seed germination, plant growth, and pollen and fruit development [38]. In the present study, malate dehydrogenase was phosphorylated at Thr-188, and the phosphorylation activity was up-regulated in the UP samples (Figure 5B). There is scarce research on MDH phosphorylation. However, the study conducted by Khan et al. [39] suggested that malate dehydrogenase is likely involved in gibberellin signaling in the rice leaf sheath and phosphorylation of MDH participates in the gibberellin-mediated elongation of rice leaves. Therefore, we hypothesized that up-regulated Thr-188 phosphorylation in UPs might enhance the activity of MDH and promote the rapid elongation of pegs after their penetration into the ground.

Phosphoenolpyruvate carboxylase (PEPC, EC 4.1.1.31, and Ser-469) and phosphoenolpyruvate carboxykinase (PEPCK, EC 4.1.1.49, Thr-68, and Ser-77) were different between the UP and AP samples in this study. PEPC and PEPCK play different roles in the synthesis of phosphoenolpyruvate (PEP). PEP, involved in the synthesis of protein and fatty acid (FA), is a high-energy metabolic intermediate that serves as an important connection among various metabolic pathways, such as glycolysis, gluconeogenesis, and organic acid metabolism. Previous studies have discovered that dephosphorylation can enhance the activity of PEPCK and facilitate the synthesis of PEP [40]. In contrast, the dephosphorylation of PEPC might result in increased sensitivity of PEPC to the inhibition of malate (Mal) and thus decrease the activity of PEPC [41]. Therefore, we speculated that dephosphorylation of PEPC and PEPCK in UPs in our study may regulate the reverse reaction between PEPC and PEPCK, avoiding the ineffective cycle and promoting the reaction toward PEP synthesis in the UP samples. The harvested peanut pod consists of two parts, namely the shell and the seed. The term commonly known as the peanut kernel refers specifically to the seed portion of the peanut pod, which is rich in plant protein, fatty acid, and carbohydrates. The protein content of peanuts ranges from 25% to 30%, only less than soybeans among legumes and higher than that of sesame, rape, and cottonseeds. The fat content of peanut seeds is as high as 43–55%, and the carbon water content is about 20%. We speculated that after the pegs undergo mechanical stimulation and dark conditions entering into the soil, they begin to provide energy and metabolites for pod development. It appears, therefore, that protein phosphorylation plays an important role in the process of peanut gynophore implantation. Phosphorylation regulates the accumulation of nutrients required for the development of subterranean peanut pods, thereby improving peanut yield.

## 4. Materials and Methods

### 4.1. Plant Materials and Paraffin Section

This study was conducted at Yinmaquan Station of Shandong Academy of Agricultural Sciences (117°5′ E, 36°43′ E, Jinan, China) using the peanut variety “Huayu 25” (provided by Shandong Peanut Research Institute) as the material. The seeds of “Huayu 25” were grown in the field. When the peanut entered the flowering period and the pegs grew, we prepared to obtain samples. The proteomic and phosphorylation groups were determined by arial pegs (APs) and underground pegs (UPs), which have been penetrated into the ground for 5 days. There were three biological replicates for both proteome and phosphorylation experiments.

The paraffin section method was used to observe the different developmental processes of the pegs. After the sample APs and UPs were fixed with FAA fixing solution, they were dehydrated, made transparent, and then dipped in wax step by step; the material was embedded in paraffin for fixation, wax blocks with a thickness of 10 μM were picked up, and then wax tape was glued to the slide to spread and dry the sheet, which was dewaxed, rehydrated, and then dyed with Tolonium chloride. Finally, it could be observed with a Nikon ECLIPSE TI-SR microscope (Tokyo, Japan). The experiment was repeated with more than 20 gynophores in each stage.

### 4.2. Protein Preparation and Mass Spectrometry Analysis

The peanut gynophores of APs and UPs were retrieved from −70 °C and were fully ground into powder with liquid nitrogen. Each treatment was repeated three times. For each group of samples, four times the volume of phenol extraction buffer was added, followed by ultrasonication. Tris–Phenol was added and centrifuged at 10,000× *g* and 4 °C for 10 min. The supernatant was collected, and then 0.1 M ammonium acetate/methanol at 5 times the volume was added prior to precipitation overnight. The protein precipitation was washed using methanol and acetone, respectively. Finally, the precipitate was treated with 8 M urea. The protein concentration was determined by a BCA kit (Shanghai, China). After digestion of trypsin, the tryptic peptides were fractionated into fractions by high-pH reverse-phase HPLC.

### 4.3. Phosphoproteomic Analysis

The proteins were decomposed into peptides with lysate. The peptide mixtures were first incubated with an IMAC microsphere (Thermo, A32992, Waltham, MA, USA) suspension with vibration in loading buffer (50% acetonitrile/0.5% acetic acid). Subsequently, they were shaken for incubation. After incubation, the material was washed with buffer solution 3 times successively. Finally, phosphopeptides were eluted with 10% ammonia water. The eluent was collected, vacuum-frozen, and then drained for liquid chromatograph–tandem mass spectrometry (LC–MS/MS) analysis.

The peptides were dissolved by liquid chromatography mobile phase A and separated using EASY-nLC 1200 ultra-high performance liquid chromatography (UPLC). Briefly, secondary mass spectrometry data were retrieved using Proteome Discoverer 2.4 (v2.4.1.15) [42]. The database was Blast_Arachis_hypogaea_3818_PR_20210319.fasta (97,596 sequences). An anti-database was incorporated to estimate the false discovery rate (FDR) due to random matches, and a common contamination database was included to mitigate the impact of contaminant proteins on the identification results. The enzyme digestion parameters were configured as Trypsin (Full), allowing for up to 2 missed cleavages, with a minimum peptide length of 6 amino acid residues. The maximum number of variable modifications per peptide was set to 3. The mass error tolerances were specified as 10 ppm for precursor ions and the mass error tolerance of secondary fragment ions was 0.02 Da. Carbamidomethyl (C) was designated as a fixed modification, while Oxidation (M), Acetyl (N-terminus), Met-loss (M), Met-loss+acetyl (M), Phospho (S, T, Y), and Deamidated (N, Q) were set as variable modifications. The FDRs for proteins, peptides, and PSM identifications were all set at 1%. In this study, quantitative values for each sample in three replicates were obtained via three complete quantitative experiments. The first step involves identifying the distinct modified phosphate groups in the two samples. The mean of each sample is computed from the three technical replicates, and, subsequently, the ratio of the means between the two samples is calculated. The income ratio serves as the final quantification. The second step is to calculate the significance (*p*-value) based on phosphorylation expression differences. *p* < 0.05 and a ratio < 1/1.5 were adopted as thresholds to determine up-regulation, while down-regulation was determined by *p* < 0.05 and a protein ratio < 0.833 as thresholds. The original abundance ratio of the phosphorylation site was normalized in accordance with the corresponding protein ratio.

### 4.4. Bioinformatics Analysis

The protein quantitative PCA results of all samples are shown in the graph, in which the degree of aggregation among samples represents the difference between samples. Common functional annotations were made for the identified proteins, including Gene Ontology (GO) (http://geneontology.org/), the Kyoto Encyclopedia of Genes and Genomes (KEGG) (http://www.genome.ad.jp/kegg/), COG/KOG (http://www.ncbi.nlm.nih.gov/COG), the protein–protein interaction network (http://string-db.org), and protein subcellular localization (http://www.genscript.com/wolf-psort.html). The Fisher exact test was used for statistical analysis of pathway enrichment, and a correction threshold *p* < 0.05 was considered as a significant enrichment pathway. As detailed in [43], enrichment-based clustering utilizes distinct protein functional classifications. As described in [44], phosphorylation motifs of DRPPs were analyzed with motif-X software. The identified proteins and phosphorylated proteins were analyzed and clustered by the K-means method.

### 4.5. Quantification of Protein Abundance and Phosphorylation Level of Selected Phosphosites by PRM

Proteins which were randomly selected from the samples were quantitatively verified by the parallel reaction monitoring (PRM) technique. The protein was extracted from the samples, digested using trypsin, and then analyzed using liquid chromatography–mass spectrometry. The specific PRM validation procedure was obtained from Jingjie PTM BioLab Co., Ltd. (Hangzhou, China) described as in [45].

### 4.6. Phos-Tag SDS-PAGE

Phos-tag^TM^, purchased from FUJIFILM Wako Pure Chemical Corporation (Tokyo, Japan), is a prefabricated gel with 50 μmol/L Phos-tag^TM^ Acrylamide, ready for use when unpacked. The prefabricated glue contains zinc as a metal ion and is stable in the central gel buffer. An SDS-PAGE experiment was performed and then transferred to the membranes, which were incubated with the first antibody anti-SPS (dilution 1:500) and second antibody successively. Finally, observations were made.

### 4.7. Detection of Starch

The samples were deoxidized at 105 °C for 40 min and then dried at 80 °C. Determination of the content of amylose and amylopectin was performed using the dual-wavelength method [46]. The dual-wavelength method is a commonly used method to measure starch content, which can calculate the starch concentration by measuring the absorbance of starch molecules at two different wavelengths.

## 5. Conclusions

In this study, phosphoproteomics was used for the first time to study the post-translational regulation of peanut gynophores before and after soil penetration. A total of 6859 proteins and 4142 unique phosphoproteins with 10,070 phosphosites were identified. Our data provide new insights into the mechanism of peanut peg insertion, suggesting that phosphorylation is involved in the rapid elongation of the pegs, along with providing energy for pod development and preparing the synthesis of metabolites during pod development after mechanical stimulation and darkness.

## Figures and Tables

**Figure 1 ijms-26-00634-f001:**
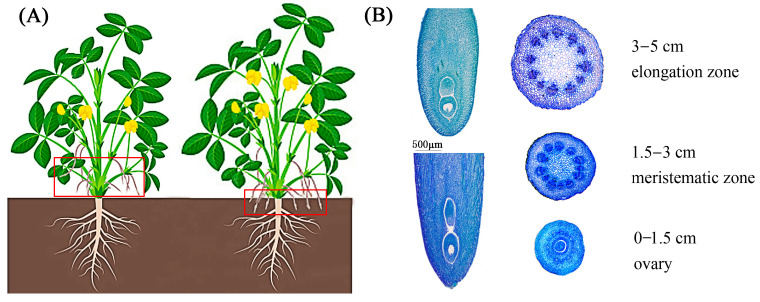
A diagram of peanut gynophore development. (**A**): The peanut gynophores that were about to be inserted into the soil and the underground peanut gynophores that had just been inserted into the soil for the assay; (**B**): paraffin section of peanut gynophores before and after implantation.

**Figure 2 ijms-26-00634-f002:**
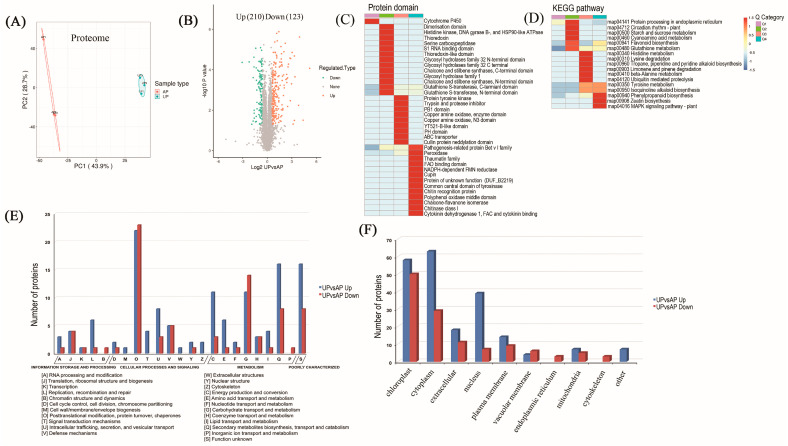
Proteome data analysis. (**A**): Scatter plot of quantitative principal component analysis between repeated samples in the proteome data; (**B**): statistical map of differential proteins; (**C**,**D**): distribution of proteins contained and KEGG analysis for differential protein enrichment; (**E**): the Clusters of Orthologous Groups of proteins (KOG/COG) classification of the DAPs between APs and UPs; (**F**): the subcellular localization number of DAPs in APs and UPs.

**Figure 3 ijms-26-00634-f003:**
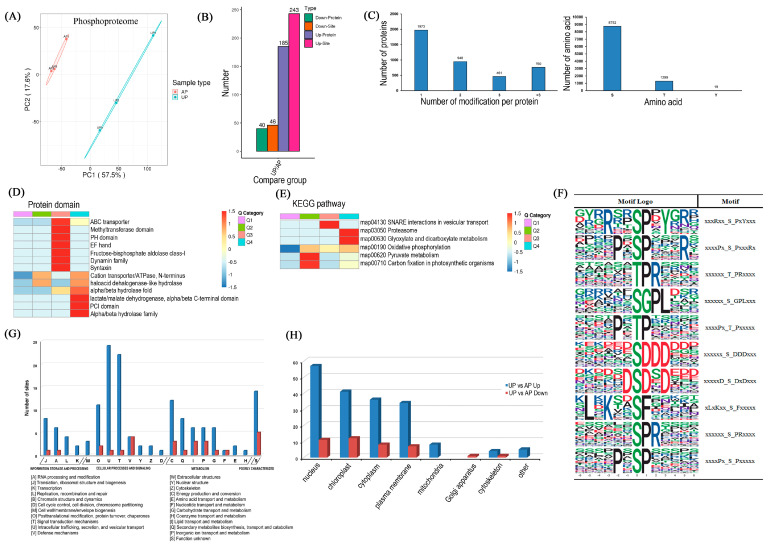
Overview of the phosphoproteome of peanut gynophores after penetrating into the soil. (**A**): Scatter plot of quantitative principal component analysis between repeated samples in the phosphoproteome data; (**B**): up- and down-regulated phosphor-sites and phosphor-proteins that were statistically significant are represented in the scatter plots; the number of proteins classified according to the number of phosphorylated sites (**C**) and amino acid classification (**D**), respectively; (**E**): KEGG analysis for DRPPs; (**F**): sequence motif analysis of phosphorylation sites; (**G**): the Clusters of Orthologous Groups of proteins (KOG/COG) classification of the DRPPs between APs and UPs; (**H**): statistics of DRPP subcellular localization in APs and UPs.

**Figure 4 ijms-26-00634-f004:**
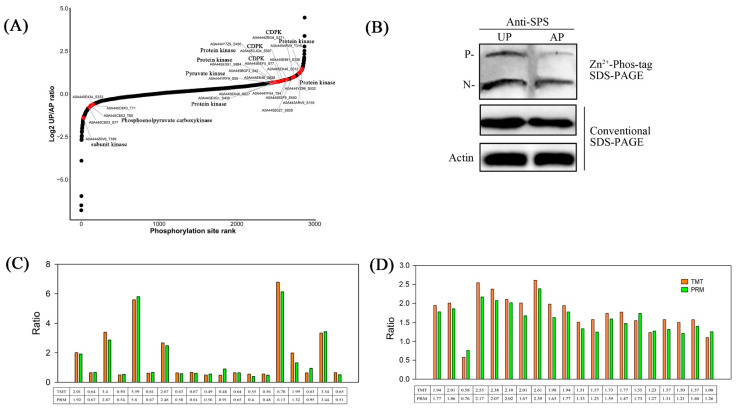
(**A**): Ratio scatter plot of phosphorylation sites. For the identified modification sites, a dot diagram was constructed after sequencing based on log2 (ratio). The dot color of the significantly different kinase phosphorylation sites (the significant difference sites of protein description encompassing kinase) is red, and a line is drawn to mark the protein accession_modification sites. (**B**): Mobility shift detection of phosphorylated and non-phosphorylated SPS protein in APs and UPs. Immunoblot analyses were performed using anti-SPS antibodies. The phosphorylated form (P) and non-phosphorylated form (N) are indicated. (**C**): Comparison between tandem mass tag labeling (TMT) for relative quantitation and parallel reaction monitoring (PRM). Proteins in PRM experiments were randomly selected from samples. (**D**): PRM assay results showing the associated pathways of the 20 phosphosites identified in UPs and APs. Each value represents the mean of three replicates.

**Figure 5 ijms-26-00634-f005:**
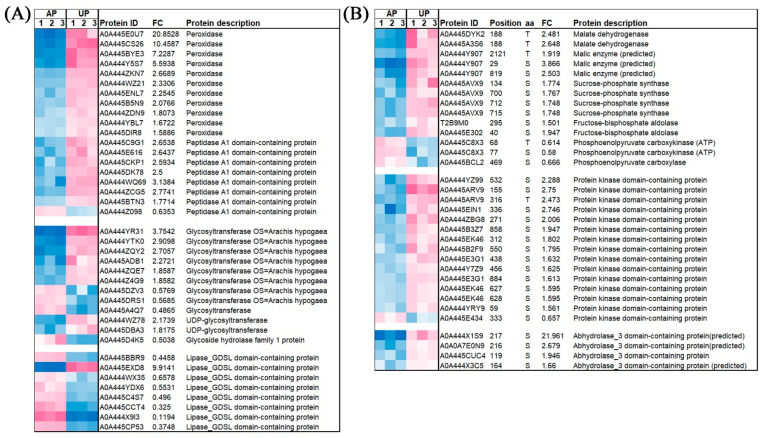
Expression analysis of some important DEGs (**A**) and DRPPs (**B**) between APs and UPs, respectively. Higher levels are shown in light red and lower levels are shown in blue.

**Figure 6 ijms-26-00634-f006:**
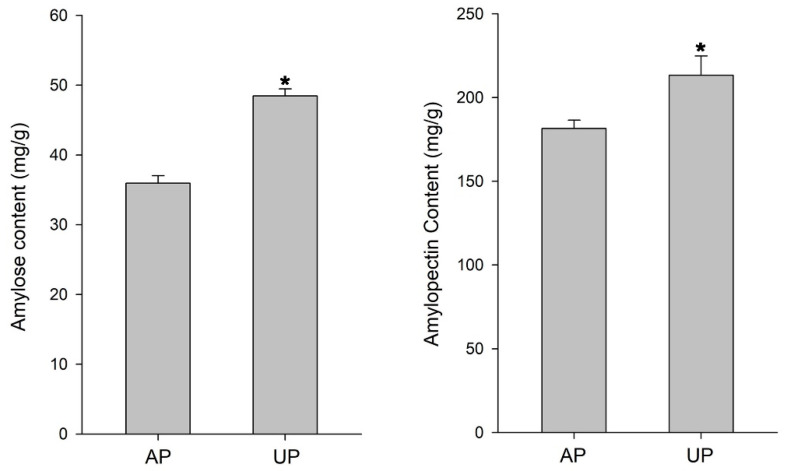
The contents of amylose and amylopectin. *p* values were calculated by using *t*-test and are indicated by asterisks (*) when significantly different from CK treatments (*p* < 0.05).

## Data Availability

The mass spectrometry proteomics data have been deposited in the ProteomeX change Consortium via the PRIDE (Perez-Riverol et al., 2019) partner repository with the dataset identifiers PXD042065 and PXD042066.

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
