# Peer review of "Deciphering the Proteome and Phosphoproteome of Peanut (Arachis hypogaea L.) Pegs Penetrating into the Soil"

_ijms, 2025, doi:10.3390/ijms26020634_

Round 1

Reviewer 1 Report

Comments and Suggestions for Authors

Dear Authors,

Your manuscript entitled 'Deciphering the Proteome and Phosphoproteome of Peanut (Arachis hypogaea L.) Pegas Penetrating into the Soil' might be suitable for publication in International Journal of Molecular Sciences after major revision. Please, see below a list of comments/suggestions to be applied by you before accepting it.

Figures 2-5 Increase full size to gain visibility. Separate graphs if necessary to improve readers' reading.

L365-L393 Give details about the experimental design applied. Tell us from when to when this experiment was conducted, how samples were collected and provide set conditions for microscope analysis and mass spectrometry. Specify which were your treatments and how many samples were attributed to each treatment. Explain how many samples were taken for análisis.

L415-416 Review this sentence. Reformulate it.

L443-L444 Complete this section.

L453-L457 Check the format. Rewrite this section.

Yours sincerely,

Reviewer.

Author Response

Dear editors,

Thank you for your letter and for the reviewers’ comments on our manuscript entitled “Deciphering the Proteome and Phosphoproteome of Peanut (Arachis hypogaea L.) Pegs Penetrating into the Soil”. These comments are very valuable and helpful for us to improve our paper. The main corrections in the paper and the response to the reviewer’s comments are as follows.

Reviewer 1:

Thank you for your questions and suggestions.

Your manuscript entitled 'Deciphering the Proteome and Phosphoproteome of Peanut (Arachis hypogaea L.) Pegas Penetrating into the Soil' might be suitable for publication in International Journal of Molecular Sciences after major revision. Please, see below a list of comments/suggestions to be applied by you before accepting it.

  1. Question (Q): Figures 2-5 Increase full size to gain visibility. Separate graphs if necessary to improve readers' reading.

Reply (R): Thank you for your suggestions. We have changed the size of Figures 2-5 and increase the size and font of each image.

  1. Question (Q): L365-L393 Give details about the experimental design applied. Tell us from when to when this experiment was conducted, how samples were collected and provide set conditions for microscope analysis and mass spectrometry. Specify which were your treatments and how many samples were attributed to each treatment. Explain how many samples were taken for análisis.

Reply (R): Thank you for your suggestion. We have added all the details in the manuscript and showed in red.

  1. Question (Q): L415-416 Review this sentence. Reformulate it.

Reply (R): We have revised the reformulate it. “Peptide mixtures were first incubated with IMAC microspheres (Thermo, A32992) suspension with vibration in loading buffer (50% acetonitrile/0.5% acetic acid)”.

  1. Question (Q): L453-L457 Check the format. Rewrite this section.

Reply (R): We are very sorry and we have rewrited this section. Funding: This study is supported by the Shandong Key R&D Program (Major Scientific and Technological Innovation Project) (ZFJH202310), the Natural Science Foundation of China (32272020, 32272227), the Taishan Scholars Program (tsqn202211275, tspd20221107, tsqn202408305), Shandong Province Key Research and development Project (2022CXPT031).

  1. Question (Q): L443-L444 Complete this section.

Reply (R): Thank you for your suggestion. We have completed this section.

Reviewer 2 Report

Comments and Suggestions for Authors

The authors identified and quantified the proteins and phosphoproteins from the peanut pegs. The manuscript needs thorough revision.

1.     The subsection “2.2. Proteomic identification of underground pegs” is unclear. The authors stated, “In the study, the total spectral number is 659649, of which the effective spectral number is 293262.” What does the effective spectral number mean? How many are proteins and phosphoproteins in each sample? The authors should provide the table with all these identifications. How is the PCA plot created? How many proteins were considered? Figure 2B is unclear and needs to be improved. The log2 FC should be >2.

2.     “The modified DAPs were compared with the STRING (v.11.0) protein interaction network database, and the protein interaction relationships were extracted according a confidence score > 0.4 threshold.” The confidence score needs to be considered between 0.7 and 0.9.

3.     The Materials and Methods section needs to be rewritten for clarity and reproducibility. How were the protein samples extracted from paraffin sections and the proteomic samples were prepared? This section needs to be organized properly in order.

4.     How phosphopeptides were obtained? The authors stated, “The enriched buffer solution (50% acetonitrile/0.5% acetic acid) dissolved the peptide, and the supernatant was transferred in advance to the IMAC material.” How could the authors enrich the buffer and not the samples?

5.     How phosphopeptides were analyzed on mass spec? What were the specific parameters? How were the data analyzed on MaxQuant and downstream steps? The authors stated, “For phosphorylation site analysis, the output of MaxQuant is utilized for the entry with the lowest available underline strength.” This is unclear.

Minor:

The authors need to check the spelling and grammar. E.g. “4. Materiels and methods” it should be Materials and Methods

Figure quality need to be improved.

Comments on the Quality of English Language

The authors need to check the spelling and grammar. E.g. “4. Materiels and methods” it should be Materials and Methods

Author Response

Dear editors,

Thank you for your letter and for the reviewers’ comments on our manuscript entitled “Transcriptomic and metabolomic analyses reveal the roles of Flavonoids and auxin on peanut nodulation”. These comments are very valuable and helpful for us to improve our paper. The main corrections in the paper and the response to the reviewer’s comments are as follows.

Reviewer 2:

Thank you for your questions and suggestions.

  1. Question (Q): The subsection “2.2. Proteomic identification of underground pegs” is unclear. The authors stated, “In the study, the total spectral number is 659649, of which the effective spectral number is 293262.” What does the effective spectral number mean? How many are proteins and phosphoproteins in each sample? The authors should provide the table with all these identifications. How is the PCA plot created? How many proteins were considered? Figure 2B is unclear and needs to be improved. The log2 FC should be >2.

Reply (R): Thank you for your suggestion. We are sorry for our inaccurate description. The effective spectral number is Matched spectrums and we have corrected it in the manuscript. The number of proteins and phosphoproteins in each sample was shown in supplementary table 1. We have changed the size of Figures 2-5 and increase the size and font of each image. The description of PCA has been added in Materiels and Methods.

  1. Question (Q): “The modified DAPs were compared with the STRING (v.11.0) protein interaction network database, and the protein interaction relationships were extracted according a confidence score > 0.4 threshold.” The confidence score needs to be considered between 0.7 and 0.9.

Reply (R): Thank you for your suggestion. We miswrote 0.7 as 0.4, and we apologize for such a simple error. We have corrected it in the manuscript.

  1. Question (Q): The Materials and Methods section needs to be rewritten for clarity and reproducibility. How were the protein samples extracted from paraffin sections and the proteomic samples were prepared? This section needs to be organized properly in order.

Reply (R): The Materials and Methods section was showed after discussion in the manuscript. The paraffin sections didn’t require protein samples. We have supplemented the method in detail in the Materials and Methods.

  1. Question (Q): How phosphopeptides were obtained? The authors stated, “The enriched buffer solution (50% acetonitrile/0.5% acetic acid) dissolved the peptide, and the supernatant was transferred in advance to the IMAC material.” How could the authors enrich the buffer and not the samples?

Reply (R): We are very sorry and we have reformulated it “Peptide mixtures were first incubated with IMAC microspheres (Thermo, A32992) suspension with vibration in loading buffer (50% acetonitrile/0.5% acetic acid)”.

  1. Question (Q): How phosphopeptides were analyzed on mass spec? What were the specific parameters? How were the data analyzed on MaxQuant and downstream steps? The authors stated, “For phosphorylation site analysis, the output of MaxQuant is utilized for the entry with the lowest available underline strength.” This is unclear.

Reply (R): Thank you for your suggestion. We are sorry for our unclear description and we have added the detail in the Phosphoproteomic analysis.

  1. Question (Q): The authors need to check the spelling and grammar. E.g. “4. Materiels and methods” it should be Materials and Methods.

Reply (R): Thank you for your suggestion. We have reviewed carefully the full artical and revised the all the figures.

Reviewer 3 Report

Comments and Suggestions for Authors

The paper takes an interesting approach to understanding a physological mechanism, such as pod formation, that only occurs in a very limited number of plants and very few crops. 

The amount of raw data is huge, but the authors draw very general and descriptive conclusions. In my opinion, such an experimental design and analysis merits a more precise conclusion and a better presentation of the results.

The conclusion says that "A total of 6859 448

We identified 4142 unique phosphoproteins and 10,070 phosphosites. Our data provide new insights into the mechanism of peanut peg insertion" and then "suggesting that phosphorylation is involved in the rapid elongation of the pegs, along with providing energy for pod development and preparing the synthesis of metabolites during pod development after mechanical stimulation and darkness."

Authors should obtain more information on the data. For instance: Is there any conserved phosphosite among proteins related to the same biological process suggesting a common protein kinase? Are the protein kinases or phosphatases identified suggesting a kinase cascade? Is there anything previously published in other plants on protein kinases related to this physiological process? Have these proteins been identified in the present study?

Results presentation:

Please enlarge figures 2, 3, and 4, as the font size is too small and nothing can be read.

Figure 6: Which is the n number for each bar? Please perform a statistical analysis to check the significance of the results.

Author Response

Dear editors,

Thank you for your letter and for the reviewers’ comments on our manuscript entitled “Transcriptomic and metabolomic analyses reveal the roles of Flavonoids and auxin on peanut nodulation”. These comments are very valuable and helpful for us to improve our paper. The main corrections in the paper and the response to the reviewer’s comments are as follows.

Reviewer 3:

Thank you for your questions and suggestions.

  1. Question (Q): Authors should obtain more information on the data. For instance: Is there any conserved phosphosite among proteins related to the same biological process suggesting a common protein kinase? Are the protein kinases or phosphatases identified suggesting a kinase cascade? Is there anything previously published in other plants on protein kinases related to this physiological process? Have these proteins been identified in the present study?

Reply (R): Thank you for your suggestion. Some conserved phosphosite such as RxxS motifs have been shown to be potential substrates for protein kinase A, mitogen-activated protein kinase (MAPKK), and calmodulin-dependent protein kinase (CDPK). The Pro-directed motifs SP and SPxxxxR, which are putatively phosphorylated by MAPKs, were found in 316 and 172 phosphorylated peptides, respectively. We have supplemented in the manuscript.

Protein phosphorylation and dephosphorylation are regulated by different kinases and phosphatases, and the protein kinases and phosphatases identified don’t necessarily indicate the presence of a kinase cascade pathway. This is the first study on post-translational regulation of peanut gynophores before and after soil penetration by phosphoproteomics. The growth of peanut is different from that of other plants, so the process of peanut gynophores (pegs) insertion has not been reported in other plants.

  1. Question (Q): Please enlarge figures 2, 3, and 4, as the font size is too small and nothing can be read.

Reply (R): Thank you for your suggestions. We have changed the size of Figures 2-5 and increase the size and font of each image.

  1. Question (Q): Figure 6: Which is the n number for each bar? Please perform a statistical analysis to check the significance of the results.

Reply (R): We have performed the statistical analysis and added in the Figure 6.

Reviewer 4 Report

Comments and Suggestions for Authors

The submitted manuscript deals with a very interesting issue that has overlap not only in theoretical research but also in practical research. The use of the knowledge gained could be used to understand the defensive reactions of plants (roots, stolons, shoots, etc.) to soil pathogens. The manuscript is written with great care and quality. It is somewhat unfortunate that the abstract is written in general terms, and it would be useful to make it somewhat more specific. The methods section describes the individual methods very well. The only thing that is somewhat lacking is information on the source of the plant material or the method of growing the plants. I think that this should be mentioned. In the results, anatomical sections of the roots are shown in the figure. It might be useful to describe them. It would also be useful to enlarge Figures 2-5. I would also recommend rechecking the review of the lithology used and unifying it (see publications 5 and 7).

Author Response

Dear editors,

Thank you for your letter and for the reviewers’ comments on our manuscript entitled “Transcriptomic and metabolomic analyses reveal the roles of Flavonoids and auxin on peanut nodulation”. These comments are very valuable and helpful for us to improve our paper. The main corrections in the paper and the response to the reviewer’s comments are as follows.

Reviewer 4:

Thank you for your questions and suggestions.

  1. Question (Q): The submitted manuscript deals with a very interesting issue that has overlap not only in theoretical research but also in practical research. The use of the knowledge gained could be used to understand the defensive reactions of plants (roots, stolons, shoots, etc.) to soil pathogens. The manuscript is written with great care and quality. It is somewhat unfortunate that the abstract is written in general terms, and it would be useful to make it somewhat more specific. The methods section describes the individual methods very well. The only thing that is somewhat lacking is information on the source of the plant material or the method of growing the plants. I think that this should be mentioned. In the results, anatomical sections of the roots are shown in the figure. It might be useful to describe them. It would also be useful to enlarge Figures 2-5. I would also recommend rechecking the review of the lithology used and unifying it (see publications 5 and 7).

Reply (R): Thank you for your suggestion. We have changed the size of Figures 2-5 and increase the size and font of each image. We have added all the details on the source of the plant material and the method of growing the plants which showed in red in the manuscript. In the results, anatomical sections of the pegs have been described in the results. We also rechecked the review of the lithology.

In our second half of the abstract “Once the pegs have successfully entered the soil, disease-resistant and stress-response proteins, such as glutathione S-transferase, peroxidase, and cytochrome P450 significantly increased in the UP samples in order to adapt to the new soil environment. The increase abundance of photosynthe-sis-associated proteins in UP samples provided more abundant photosynthetic products, which provided the preparation for the subsequent pod development. Phosphoproteomics reveals the regulatory network of the synthesis of nutrients such as starch, protein and fatty acid (FA). These results provide new insights into the mechanism indicated that after the pegs were inserted into the soil, phosphorylation is involved in the rapid elongation of pegs and accompany supplying energy for pod development and preparing for the synthesis of metabolites during pod development following mechanical stimulation and darkness” has been summarized the main content of the paper.

Round 2

Reviewer 1 Report

Comments and Suggestions for Authors

Dear Authors,

After your revision your manuscript might be now considered suitable for publication in IJMS in present form.

Yours sincerely,

Reviewer.

Author Response

Thank you very much for your recognition.

Reviewer 2 Report

Comments and Suggestions for Authors

The revised version of the manuscript does not seem to be improved and needs thorough revision.

1.     The authors should provide a table or bar graph with all the proteins, peptides, and phosphopeptides identified and quantified in each sample. The table in supporting information is not helpful.

2.     There is no improvement in Figure 2B. In Figure 4A, indicate the gene name instead of protein accession.

3.     There is no change “4. Materiels and methods” should be Materials and Methods

4.     The authors made minor changes in the main text, but it did not improve the overall quality. The whole section needs to be rewritten.  There are several incidences where the authors made duplicate statements. E.g. “After digestion of trypsin, the peptides were dissolved by liquid chromatography mobile phase A and separated using the EASY-nLC 1200 ultra-high performance liquid phase system. High-resolution Orbitrap was used to detect and analyze the peptides and their secondary fragments. The peptides were separated by an ultra-high performance liquid phase system and injected into an NSI ion source for ionization and then analyzed by Orbitrap Exploris™ 480 mass spectrometry”

5.     The whole Methods section needs to be reorganized. For subsection 4.5. Phosphoproteomic analysis from where the Peptide mixtures came?

6.     The authors stated, “Tandem mass spectra were blasted against…” This is not the correct sentence. The authors used MaxQuant before and are now stating they used the Mascot search engine (v.2.3.0). This creates more ambiguity.

Comments on the Quality of English Language

The authors need to check the spelling and grammar. Also, improve the writing style.

Author Response

Dear editors,

Thank you for your letter and for the reviewers’ comments on our manuscript entitled “Deciphering the Proteome and Phosphoproteome of Peanut (Arachis hypogaea L.) Pegs Penetrating into the Soil”. These comments are very valuable and helpful for us to improve our paper. The main corrections in the paper and the response to the reviewer’s comments are as follows.

Reviewer 2:

Thank you for your questions and suggestions.

  1. Question (Q): The authors should provide a table or bar graph with all the proteins, peptides, and phosphopeptides identified and quantified in each sample. The table in supporting information is not helpful.

Reply (R): Thank you for your suggestion. We have provided a table with all the proteins, peptides, and phosphopeptides identified and quantified in each sample in supplementary table S1.

  1. Question (Q): “There is no improvement in Figure 2B. In Figure 4A, indicate the gene name instead of protein accession.

Reply (R): Thank you for your suggestion. We have tried our best to enlarge the size and font of Figure 2 and in Figure 2B Volcano maps are a way of counting. In the Figure 4A, we have added the name of some important proteins. Because some of these proteins are repetitive.

  1. Question (Q): There is no change “4. Materiels and methods” should be Materials and Methods.

Reply (R): We are sorry for the mistakes. We only changed “methods” to “Methods” and we have used Materials and Methods in the manuscript.

  1. Question (Q): The authors made minor changes in the main text, but it did not improve the overall quality. The whole section needs to be rewritten.  There are several incidences where the authors made duplicate statements. E.g. “After digestion of trypsin, the peptides were dissolved by liquid chromatography mobile phase A and separated using the EASY-nLC 1200 ultra-high performance liquid phase system. High-resolution Orbitrap was used to detect and analyze the peptides and their secondary fragments. The peptides were separated by an ultra-high performance liquid phase system and injected into an NSI ion source for ionization and then analyzed by Orbitrap Exploris™ 480 mass spectrometry”

Reply (R): We are very sorry and we have reformulated it “The peptides separated by an ultra-high performance liquid phase system were injected into an NSI ion source for ionization and then analyzed by Orbitrap Exploris™ 480 mass spectrometry”. We have checked the whole manuscript and improved the overall quality.

  1. Question (Q): The whole Methods section needs to be reorganized. For subsection 4.5. Phosphoproteomic analysis from where the Peptide mixtures came?

Reply (R): Thank you for your suggestion. We are sorry for our unclear description and we have added the details in the subsection 4.5.

  1. Question (Q): The authors stated, “Tandem mass spectra were blasted against…” This is not the correct sentence. The authors used MaxQuant before and are now stating they used the Mascot search engine (v.2.3.0). This creates more ambiguity.

Reply (R): Thank you for your suggestion. Actually, we have complemented the analysis method MS/MS data. Mascot is a powerful search engine for the identification of proteins and peptides through mass spectrometry data. We used it for qualitative analysis of differential proteins. MaxQuant is a comprehensive quantitative mass spectrometry software platform for proteomic quantification and mass spectrometry data management. The MaxQuant output file (“Phospho (STY) sites.txt”) were used to quantify phos-phorylation sites.

Round 3

Reviewer 2 Report

Comments and Suggestions for Authors

The revised version is not much improved.

As I read the manuscript again and again, I found the whole methods section is ambiguous.

For example- The authors stated, "Briefly, the data were 417 processed using Mascot search engine (v.2.3.0). Tandem mass spectra were blasted against 418 Uniprot_foxtail_4555 (http://www.uniprot.org/taxonomy/4555) database concatenated 419 with reverse decoy database. The mass tolerance for precursor ions was set as 20 ppm in 420 First search and 5 ppm in Main search, and the mass tolerance for fragment ions that set 421 as 0.02 Da. FDR was adjusted to < 1%, and the minimum score for modified peptides was 422 set to > 40. For phosphorylation site analysis, the lowest available underscore intensity 423 entries from the MaxQuant output were used."

Here the authors stated Mascot search and used MaxQuant output. How this is possible? Mascot and MaxQuant are different software and has different capabilities. 

The authors stated, "The resulting MS/MS data were processed according to [42]." It is not clear how the data were acquired on mass spec. The citing reference is not for phosphoproteins/peptides, it is for metabolites. 

The authors stated they used Phosphosite (STY) from MaxQuant but Figure 3F shows the other residues enriched which is wrong. 

These are three examples, but there are several examples in the entire manuscript that need to be addressed. 

Comments on the Quality of English Language

The writing style needs to improve. 

Author Response

Dear editors,

Thank you for your letter and for the reviewers’ comments on our manuscript entitled “Deciphering the Proteome and Phosphoproteome of Peanut (Arachis hypogaea L.) Pegs Penetrating into the Soil”. These comments are very valuable and helpful for us to improve our paper. The main corrections in the paper and the response to the reviewer’s comments are as follows.

Reviewer 2:

Thank you for your questions and suggestions.

  1. Question (Q): Here the authors stated Mascot search and used MaxQuant output. How this is possible? Mascot and MaxQuant are different software and have different capabilities.

Reply (R): We are sorry for the mistakes. we have modified the analysis method for MS/MS data after communicating with the sequencing company.

  1. Question (Q): The authors stated, "The resulting MS/MS data were processed according to [42]." It is not clear how the data were acquired on mass spec. The citing reference is not for phosphoproteins/peptides, it is for metabolites.

Reply (R): Thank you for your suggestion. I have illustrated the analysis method for MS/MS data in the manuscript. We are sorry for the mistake and thank you for your help.

  1. Question (Q): The authors stated they used Phosphosite (STY) from MaxQuant but Figure 3F shows the other residues enriched which is wrong.

Reply (R): We are sorry for the mistakes. We have carefully reviewed and modified the entire method section in the manuscript.

Round 4

Reviewer 2 Report

Comments and Suggestions for Authors

The manuscript can be accepted.